# The Mental Health of the Peruvian Older Adult during the COVID-19 Pandemic

**DOI:** 10.3390/ijerph192416893

**Published:** 2022-12-15

**Authors:** Jack Roberto Silva Fhon, Maritza Evangelina Villanueva-Benites, Maria del Pilar Goméz-Luján, Maria Rosario Mocarro-Aguilar, Orfelina Arpasi-Quispe, Reyna Ysmelia Peralta-Gómez, Sofia Sabina Lavado-Huarcaya, Zoila Esperanza Leitón-Espinoza

**Affiliations:** 1Medical-Surgical Department, Nursing School, University of São Paulo, São Paulo 05403, Brazil; 2Department of Clinical Sciences, Faculty of Nursing, National University of the Peruvian Amazon, Maynas 16001, Peru; 3Faculty of Nursing, National University of Trujillo, Juan Pablo II Av, Trujillo 13001, Peru; 4Professional School of Nursing, Norbert Wiener University, Lima 15046, Peru; 5Postgradute School, Peruvian Union University, Lima 15464, Peru; 6Faculty of Nursing, National University of San Agustin of Arequipa, Arequipa 04001, Peru; 7University Social Responsibility Network of the Organization, Santo Toribio de Mogrovejo Catholic University, Chiclayo 14001, Peru

**Keywords:** elderly, mental health, pandemics, Peru, geriatrics

## Abstract

During the pandemic, the elderly population was the most exposed to disease and changes in their daily lives. The objective was to determine the association between demographic factors, access to health services, sources of information, and physical symptoms in the mental health of the elderly during the COVID-19 pandemic—a study with 3828 older adults residing in nine cities in Peru. The data was collected using a web-based survey, and the instruments of demographic data; exposure to information (radio, television, and internet); and presence of physical symptoms, anxiety, and perceived stress were used. Descriptive and analytical analysis was performed. Female sex, those aged between 60 and 79 years old, those with secondary education, those in their own home, those residing in an urban area, and those using public services of health predominated in this study. Likewise, 62.9% presented depressive symptoms; on the stress scale, an average of 27.81 (SD = 8.71), and on the anxiety scale, an average of 27.24 (SD = 6.04). Moreover, 65.1% reported fatigue, 62.2% had a headache, and 61.2% lack of energy. There is an association between demographic variables and the physical and psychological symptoms of stress, anxiety, and depressive symptoms in the elderly during the pandemic.

## 1. Introduction

Throughout the world, SARS-Cov-2 has caused COVID-19, which has given rise to innumerable infections and deaths; for this reason, the World Health Organization (WHO) classifies it as a pandemic [1].

In Peru, by Supreme Decree 184-2020, a state of national emergency was declared due to the severe circumstances that affected the lives of people as a result of COVID-19, and mandatory measures were established for all citizens, restricting various rights, including freedom, personal security, inviolability of domicile, freedom of assembly, and transit in the territory [2].

The first registered case was on 7 March 2020, and the first two deaths were reported on the 20th of the same month. Current statistics (up to 24 November 2022) report 217,399 deaths, with an accumulated fatality of 5.5 [3].

The Peruvian executive power, at the beginning of the pandemic, promulgated Supreme Decree No. 044-2020 on 15 March 2020, declaring a state of national emergency due to the COVID-19 outbreak **[4]**, in response to this health emergency, various measures are available, aimed at mandatory social immobilization in this health emergency. Complementing these provisions, Supreme Decree No. 184-2020-PCM requires the mandatory use of a KN95 mask or, failing that, a three-fold surgical mask to circulate on the road for public use and in closed places, as well as social distancing, no less than one meter, and frequent hand washing, among others [2,5].

Social confinement was one of the restrictive sanitary measures adopted and implemented by various countries, based on WHO recommendations, as a preventive measure against the COVID-19 pandemic. The impact of prolonged social confinement on the population, especially older adults, has generated increased stress, anxiety, and depression and even some cases of suicide [6].

Chávez-Negrete found a statistically significant relationship due to social confinement with anxiety and depression; 75% of the patients presented one of the two conditions, other confinements are an increasingly sedentary lifestyle, modification of eating habits, and adoption of harmful practices, increasing risk factors for comorbidities in older adults [7].

Due to the distancing necessary in response to the pandemic caused by COVID-19, these restrictions and limitations on social contact required the population, and older adults, in particular, to adapt to a new social/family coexistence [6].

The restrictions and social distancing were accompanied by significant information and decreased social contact, leading to a reduction in people’s well-being [8], which brought consequences for people’s mental health, such as depression, anxiety, poor quality of sleep, as well as increased use of anxiolytics, stress, obsessive compulsion, or phobic anxiety [9]. Older adults, considered the most vulnerable, were the most affected due to a break in their social circle, feelings of fragility, loneliness, stress, and irritation [10].

The population of the world displayed changes in their mental health during the pandemic; an indication is the presence of depression, which in 2017 was 3.44%, and that there was an increase in cases, seven times more, during the initial outbreak of the Covid- 19 between January and May 2020 [11]. In addition, a review identified that the prevalence of anxiety was 35% and stress was 53% in a population of 113,285 people and that these rates were higher during the COVID-19 pandemic [12], which will demand greater attention from mental health services.

During the pandemic, studies have been developed to understand the characteristics of the virus, epidemiological aspects, and the different public health measures. However, there is still little research on the mental health repercussions in the elderly.

This study may provide valuable information to experts for the prevention and control of risk factors and the planning of mental health care programs for the elderly. In that sense, this study aimed to determine the association between demographic factors, access to health services, sources of information, and physical symptoms in the mental health of the elderly during the COVID-19 pandemic.

## 2. Materials and Method

### 2.1. The Survey

Quantitative, descriptive, and cross-sectional study that follows the recommendations of the guide *Strengthening the Reporting of Observational Studies in Epidemiology* (STROBE) [13] and *Checklist for Reporting Results of Internet E-Surveys* (CHERRIES) [14].

The research was carried out in the main cities of three geographical regions of Peru, on the coast: Tumbes, Trujillo, Chiclayo, Lima, Arequipa, Tacna; from the Sierra: Cerro de Pasco, Huánuco; in the jungle: Iquitos.

The information collection was carried out between April to September 2021 through an online interview, which was disclosed on different social networks and sent to older adults via applications, such as Facebook and WhatsApp.

### 2.2. Participants

The study population consisted of 3,497,570 older adults living in Peru, according to the National Census 2017 of the National Institute of Statistics and Informatics [15]. The formula of proportions for finite populations was used to calculate the sample size, using a sampling error of 5% and a confidence level of 95%, with the final sample of 3828 older adults being the one used for convenience.

In order to participate in the study, the following inclusion criteria were applied: age equal to or greater than 60 years, access to the internet using social networks and a computer or smartphone, and answering the instrument in its entirety. The exclusion criteria were: older adults who did not finish answering the questionnaire.

The information was collected through a web-based survey sent through a link to older adults who were first directed to the digital informed consent, which they could read through and either accept or not participate in the study.

Before the collection, a pilot test was carried out to identify deficiencies and improve the instrument’s functionality, containing a drop-down system with mandatory questions. In addition, the participant could modify their answers at any time.

In order to reach the number of participants, a snowball strategy was used to send the link to other older adults indicated by the participants. The filling of the instrument lasted, on average, 25 min.

The study variables were: sex (male and female); age (in years); marital status (with and without a partner); education (no instruction, elementary, high, technical, and university); number of children; housing (own, family, rented, and others); the area where they reside (urban or rural); use of health services; and exposure to information about the pandemic through the use of the internet, TV, and radio (in hours). Likewise, physical symptoms self-reported by the older adult during the pandemic were identified.

In order to identify depressive symptoms, anxiety, and stress in the elderly, the Geriatric Depression Scale, the Geriatric Anxiety Inventory, and the Perceived Stress Scale were applied.

#### 2.2.1. Demographic Profile and Infodemic

The considered variables were sex, age, education, marital status, number of children, housing, area of residence, and use of health services. The exposure time older adults spent learning about the pandemic using TV, the internet, and the radio was collected to identify the infodemic.

#### 2.2.2. Self-Reported Physical Symptoms

Based on the impact that a person may suffer from fear [16,17], we investigated: tiredness, headache, lack of energy, muscle aches, trouble sleeping, dry mouth, chest tightness, cold sweats or chills, nutritional problems, palpitations, difficulty breathing, tremors, and decreased libido.

#### 2.2.3. Perceived Stress Scale

This measures the degree to which people evaluate situations in everyday life that can be considered stressful. It comprises 14 items and assesses the degree to which people perceive life as unpredictable, uncontrollable, or overloaded. This scale does not have a cut-off point, which means that the higher the score, the greater the perception of stress [18].

#### 2.2.4. Geriatric Depression Scale

This evaluates the presence of cognitive depressive symptoms, such as mood, hope, death wishes, and capacity for enjoyment. The scale has 15 dichotomous questions with a score from zero to 15 and a cut-off point greater than or equal to 5 points, considering participants with depressive symptoms [19].

#### 2.2.5. The Geriatric Anxiety Inventory

A self-report measure explicitly designed to be used with older adults measures three dimensions: cognitive, arousal-related, and somatic symptoms; it does not present a cut-off point, and the higher the score, the greater the anxiety [20].

### 2.3. Statistical Analysis

The information analysis was carried out using Statistical Package for the Social Sciences—SPSS v. 25 (IBM Corp., Armonk, NY, USA). Descriptive statistics were used, categorical variables were expressed through frequencies and percentages, and numerical variables through measures of central tendency and dispersion.

The Student’s *t*-test was applied in the bivariate analysis to compare the means between the dependent variables (anxiety, perceived stress, and depressive symptoms) and the independent ones (sex, age, and marital status). Pearson’s correlation test was used for numerical variables, with the dependent variables being the anxiety, perceived stress, and depressive symptoms, and the independent variables being the hours of use (internet, TV, and radio).

For the final analysis, the totals of the anxiety, perceived stress, and depressive symptoms scales were dichotomized, and the quartile method was used, with 75 being the cut-off to have better fidelity in the analysis of the information. Multiple logistic regression was used, and demographic variables, infodemic, and self-reported physical symptoms were considered independent. In all the statistical tests, the standard significance level was 0.05, with a 95% confidence interval.

This study was submitted to the Ethics Committee of the Universidad Peruana Unión (UPEU) for the approval of the informed consent with the number 2021-CE-EPG-000003, according to the Declaration of Helsinki of the World Medical Association (WMA) on ethical principles for medical research in human beings carried out in Seoul, Korea, October 2008 and the Nuremberg Declaration.

When accessing the research link, older adults read the study’s objective, had time to fill out the instrument, and had the option to accept or not participate in the study. Accepting to participate was automatically registered in a database that will be in the custody of the study coordinator for a maximum period of five years.

## 3. Results

### Descriptive Statistics

Of the 4368 participants, a predominance of the female sex was identified (56%), as well as a predominance of those between 60 and 79 years old (86.3%), with high-school education (29.8%), with their own house (80.1%), residing in an urban area (77.2%), and who make use of public health services (41%) (Table 1).

Concerning the hours of exposure that the older adults had to information about the pandemic, it was identified that the average use of the internet was 1.32 (DE = 2.52), TV was 2.16 (DE = 2.42), and radio was 1.79 (DE = 2.56).

Regarding the information on the mental health assessment of older adults during the pandemic, 2747 (62.9%) were found to have depressive symptoms. The participants presented an average of 27.81 (SD = 8.71) points on the stress scale and 27.24 (SD = 6.04) on the anxiety scale.

Among the physical symptoms that older adults presented during the pandemic are tiredness (65.1%), headache (62.2%), lack of energy (61.2%), muscle pain (60.5%), sleep problems (58.3%), and digestive problems (50.6%) (Table 2).

The mean comparison analysis identified that the variables sex, age, and marital status presented statistical significance with depressive symptoms, anxiety, and stress (Table 3).

It was observed that the hours of exposure to the news about the COVID-19 pandemic on the internet presented statistical significance with stress (*p* = 0.003). Correlation and significance were also identified between hours of television exposure with anxiety (*p* = 0.03) and stress (*p* = 0.03). In addition, radio use for hours was significant in the presence of depressive symptoms (*p* = 0.03) (Table 4).

The regression analysis observed that anxiety was associated with the female sex, the number of children, dry mouth, lack of energy, chest tightness, difficulty breathing, tremors, nutritional problems, palpitations, and fatigue.

On the other hand, perceived stress was associated with those without a partner, exposure to information about the pandemic using social media and radio, dry mouth, lack of energy, chest tightness and difficulty breathing, tremors, muscle aches, palpitations, and tiredness.

Likewise, the presence of depressive symptoms was associated with cold sweats and chills, dry mouth, lack of energy, chest tightness, and difficulty breathing. It was verified that the only variable associated with anxiety, stress, and depressive symptoms were sleeping problems (Table 5).

## 4. Discussion

In the present study, it was found that the feeling of anxiety, stress, and the presence of depressive symptoms in the elderly was associated with sociodemographic variables and some physical symptoms.

A predominance of females aged between 60 and 79 and with a partner was identified; similar results were recorded in an international study [21]. In addition, the presence of depressive symptoms, anxiety, and stress was observed. Similar results were found in a Chinese study that reported 54% moderate-to-severe psychological impact, 29% anxiety, and 8% stress with high psychological distress among participants [22].

It was verified that older age was associated with anxiety and depressive symptoms, similar to a Spanish [23] and Cuban [24] studies; this association is likely due to the perception of having lived a long time, living through a pandemic, and the proximity of death itself [22]. The awareness of their vulnerability and uncertainty [25] and the exposition of inaccurate, confusing, and contradictory information [26] increase anxiety and depression likelihood.

In this study, no relationship was found between marital status with anxiety and depression; however, in terms of stress, an OR of 0.26 was recorded with a *p* = 0.02. In this regard, an online study on pandemics was carried out in Canada about COVID-19 and the influence of marital status on stress, anxiety, and depression, with a response rate of 19.4%. Of 8267 respondents, 5799 (70.1%) identified as married, cohabiting, or with a partner, 618 (7.5%) identified as separated or divorced, 136 (1.6%) as widowed, 1541 (18.6%) as single, 95 (1.1%) as “other”, and 78 (0.9%) did not disclose their marital status. They were predominantly married, cohabiting, or with a partner (*n* = 5799, 70.1%), and had their own home (*n* = 5277, 65.7%). The average score found in all respondents was 20.79 points [27].

The average number of hours of internet use was 1.32 and radio use was 1.79 h per day, which was associated with stress. The findings demonstrate that their use generates fear and exaggeration with an overflow of sensational and fake news [28]. Digital inclusion and virtual social networks can contribute to improving communication, knowledge, leisure, cognitive stimulation, and a change in the perspective of isolation; they even promote well-being and quality of life, reducing depression [22]. Likewise, the presence of depressive symptoms was associated with cold sweats and chills, dry mouth, lack of energy, chest tightness, and difficulty breathing. It was verified that the only variable associated with anxiety, stress, and depressive symptoms was sleeping problems.

De la Serna [29] states that, among the effects of depression, there are feelings of guilt, hopelessness, and uselessness, as well as negative thoughts, in addition to an increase in pain sensitivity, persistent discomfort, digestive problems, fatigue, irritability, loss of interest in what you used to like, difficulty concentrating, and sleep disturbances, which can affect both excess and deficiency.

Having a headache is associated with anxiety; in older adults, it is characterized by recurrent, mild-to-moderate intensity, with a feeling of oppression or tension; it is not aggravated by regular physical activity. Moreover, in most cases, it has an essential association with the appearance of depressive symptoms and anxiety [23]. It seems likely that peripheral mechanisms are responsible for its genesis [30]. It is essential to know that 9% of primary headaches are not associated with organic dysfunction, including migraine, tension headaches, and trigeminal autonomic headaches. Headaches of tension origin are frequently associated with the appearance of symptoms of depression and anxiety [31].

Another physical symptom was cold sweats or chills associated with depressive symptoms. This sensation in the elderly can be produced by intense emotions accompanied by fear and terror. Somatic symptoms, low self-esteem, feelings of worthlessness, dysphoric mood, sleep disturbances, sweating, and decreased appetite mainly accompany depression [32].

In addition, it was identified that presenting depressive symptoms was associated with the sensation of having a dry mouth, which does not seem to be directly related to age but to suffering from different diseases or drugs that are usually consumed under the medical prescription stage of life. However, it can also be due to unspecific causes and suffering from stress, anxiety, or severe depression [33].

Other symptoms, such as tremors and muscle pain, were associated with anxiety and stress. The presence of depressive and anxiety disorders can cause painful manifestations, such as fibromyalgia, neuropathic pain, chronic low back pain, migraine, and painful physical symptoms, which can be persistent [34].

Likewise, chest tightness, shortness of breath, and palpitations were associated with anxiety and depression, especially in older adults. The appearance of these symptoms may be related to personality, which may be type A: competitive, restless, with a high level of stress and anxiety; type B: a calm individual with a peaceful mind, governed by values of cooperation and creativity and can be equally effective in their tasks, in that sense, those of type A are more likely to suffer heart attacks than those of personality type B [35].

Nutritional problems were identified in those who presented anxiety and depressive symptoms. Seclusion and physical inactivity force many to take harmful nutrition behaviors generating changes in their body composition, producing loss of muscle mass, and greater activation of systemic inflammation and antioxidant defenses [36]. In addition, it modified eating habits characterized by a decrease in necessary foods, such as the consumption of fruits, vegetables, and cereals. On the other hand, increased consumption of foods with excess carbohydrates, such as bread, sweets, and sugary and alcoholic beverages, generates a negative impact on the nutritional habits necessary for the body, increasing susceptibility to COVID-19 and its recovery, which can be influenced by anxiety, leading to an alteration in the self-perception of weight gain [37].

Thus, suffering from depressive symptoms causes the person to deteriorate little by little, both in terms of personal hygiene and nutrition, increasing their caloric meals intake and alcohol consumption, which will have a direct effect on weight change and over time can lead to obesity. Depression can also cause the opposite effect, a “bad” diet can lead to weight loss, which, together with the loss of sleep and being awake for more hours of the day, is characteristic of people with depression. It has been related to one of the explanations for losing weight; when a person is active, the body consumes more calories that are not replaced due to the lack of adequate nutrition [38].

The participating people shared their concerns about being overweight due to confinement. Boredom, inactivity, and bakery activities with children were the leading causes of weight gain [39]. The obese subjects presented weight gain related to eating between meals, and the subjects who exercised reported less weight gain [40].

During the quarantine, it was a source of the frustration associated with anxiety, generating changes in the choice of food and the quality of what is consumed and directing the majority of the population towards a positive energy balance. Having inadequate basic supplies, such as food, water, and clothing, among others, generated anger for 4 to 6 months after their release [41].

According to the National Institute of Statistics and Informatics to Peru [42], before the pandemic, in 20.4% of households made up of older adults, there was at least one person aged 60 and over with a caloric deficit in Peru; with this statistic being higher in metropolitan Lima (24.3%). This is followed by the rest of the urban and rural areas with 20.1% and 15.4%, respectively.

The numbers presented post-pandemic show that in 23.6% of households made up of older adults, there is at least one person aged 60 and over with a caloric deficit, with this statistic being higher in metropolitan Lima (32.8%). This is followed by the rest of the urban and rural areas with 21.7% and 13.5%, respectively. As can be seen, by the middle of 2022, there was a significant increase in older adults with a caloric deficit, with most of the affected being those who reside in metropolitan Lima [43].

Of the households headed by an older adult, 51.5% reside in a home that uses gas for cooking. In the urban area, coverage reaches 64.6% of households, while in the rural area, the percentage with an older adult head of the household who uses gas for cooking is barely 8.4% of households, which reveals a fairly marked gap between both areas (56.2 percentage points) [43].

The Ministry of Development and Social Inclusion sends a bimonthly subsidy of PEN 250 to the population in extreme poverty and poverty in the country. The same ministry, through the postcard Evidencia MIDIS, annually carries out the perception survey of the users of the Pensión 65 Program, where the results of the evaluation of the Pension 65 program users conclude that they perceive their well-being situation has improved since they are part of the Program. In 2018, it was 81.8%; in 2019, it was 80.1%; in 2020, it was 81.7%, and in the 2021, it was 73.3%. On the other hand, the amount of this subsidy spent on food in 2018 was 83.4%, in 2019 was 83.8%, in 2020 was 72.0%, and in 2021 was 80.6% [44].

On the other hand, the population in extreme poverty in Peru receives a subsidy of PEN 250 every two months, and the results of this study reflect that the non-contributory pension (Pension 65) implemented by the Peruvian State shows improvements in some indicators of the welfare of the beneficiary households, mainly in consumption spending. Likewise, it is logical to deduce that not only those affiliated (people aged 65 and over) benefited from the program but also other household members through consumption [45].

The results of the double difference model with propensity score matching (DD-PSM) show a positive impact of the program on the increase in per capita spending on food in beneficiary households. The beneficiaries of the Pension 65 program have managed to increase their per capita spending on food by 15.02%, and the result is statistically significant at a level of 10%. This result assumes that the program members are not allocating most of the subsidy to food spending. In conclusion, the Pension 65 program is an effective public policy to improve social welfare in old age, mainly through an increase in household consumption, which contributes to reducing extreme poverty in the country [45].

Regarding the fatigue symptom identified in this study, the Center for Sociological Research of Spain reported that Spanish adults during the pandemic reported feeling tired, having had sweating, tachycardia, and palpitations [46]. Anxiety, stress, and depressive symptoms can cause sleep problems due to high fear and concern about the new coronavirus [38].

Although more than half of the Peruvian sample presented this issue, this figure is slightly lower than a study in Paraguay, which indicated that 62.5% of participants reported some insomnia during social confinement [47]. Among the risk factors for insomnia in the elderly include fear of contagion, stress, and anxiety; reduced ability to cope with stress; reduced social support networks; and decreased participation in daily social activities caused by insomnia, which have all been described during the pandemic.

Disrupted cortical networks and dysregulation in the autonomic nervous system and the hypothalamic–pituitary–adrenal axis have been detected during insomnia. In addition, the hyperarousal causes increased metabolism and releases cortisol, adrenocorticotropic hormone, and others [48]. In this context, older adults would be vulnerable, so studying the temporality of these mental health problems is necessary.

The study’s main limitation is data collection; due to the need for social isolation imposed by the pandemic, the data were collected through social networks that possibly favored the population with a higher level of education and access to technology, as well as better economic resources. Another limitation of the study was using an instrument translated into Spanish but not into Peruvian Spanish, due to being part of a multicenter study and all participating countries using the same instrument. Older adults are vulnerable to COVID-19 disease, affecting their mental and physical health.

## 5. Conclusions

This study highlights the association between some demographic variables, physical symptoms, and adverse psychological events, such as stress, anxiety, and depressive symptoms, in older Peruvian adults during the COVID-19 pandemic.

It is emphasized that insomnia was one of the main manifestations reported by most participants who lived through a time of prolonged restrictive measures, such as confinement and social distancing, an unfavorable situation for the mental health of older adults.

Given the vulnerability of this population group, it is suggested that deepening the study of these variables for the design of public policies and the generation of multidisciplinary intervention strategies could improve the mental health and quality of life of the elderly.

## Figures and Tables

**Table 1 ijerph-19-16893-t001:** Sociodemographic characteristics of older adults during the COVID-19 pandemic. Peru, 2021.

Variable	Category	*n*	%	Mean (=SD *)
Sex	Female	2447	56.0	
	Male	1921	44.0	
Age	60 to 79 years	3768	86.3	69.84 (7.84)
	80 years or more	600	13.7	
Marital Status	With partner	3947	90.4	
	No partner	421	9.6	
Education	No Instruction	355	8.1	
	Elementary school	1130	25.9	
	High school	1301	29.8	
	Technical	684	15.7	
	University	898	20.6	
Number of children				4.08 (2.43)
Housing	Own	3499	80.1	
	Family	578	13.2	
	Rented	256	5.9	
	Other	35	0.8	
Residence area	Urban	3372	77.2	
	Rural	996	22.8	
Use of health services	Public	1791	41.0	
	Private	1225	28.0	
	Both	1106	25.3	
	None	246	5.6	

* SD = Standard deviation.

**Table 2 ijerph-19-16893-t002:** Self-reported physical symptoms presented by the older adult during the COVID-19 pandemic. Peru, 2021.

Symptoms	*n*	%
Fatigue	2842	65.1
Headache	2719	62.2
Lack of energy	2674	61.2
Muscle pains	2641	60.5
Sleeping problems	2546	58.3
Digestive problems	2210	50.6
Dry mouth	2105	48.2
Chest tightness	2089	47.8
Cold sweat or chills	2072	47.4
Nutritional problems	2057	47.1
Palpitations	1943	44.5
Difficulty breathing	1788	40.9
Tremors	1743	39.9
Decreased sexual libido	1316	30.1

**Table 3 ijerph-19-16893-t003:** Comparison of means of depressive symptoms, anxiety, and stress according to sociodemographic variables of the elderly during the COVID-19 pandemic. Peru, 2021.

		Depressive Symptoms		Anxiety		Stress	
Variable	Category	Mean (=SD *)	*p* †	Media (SD)	*p*	Media (SD)	*p*
Sex	Female	6.14 (3.28)	0.03	27.58 (6.12)	<0.001	28.04 (8.81)	0.05
	Male	5.93 (3.40)		26.80 (5.91)		27.523 (8.57)	
Age	60–79	5.93 (3.31)	<0.001	27.19 (5.98)	0.20	28.04 (8.51)	<0.001
	80 and over	6.80 (3.43)		27.53 (6.39)		26.37 (9.72)	
Marital Status	With partner	6.09 (3.35)	0.006	27.35 (6.05)	<0.001	27.74 (8.69)	0.09
	No partner	5.62 (3.22)		26.22 (5.91)		28.50 (8.90)	

* SD = Standard deviation; † *p* = *p* value.

**Table 4 ijerph-19-16893-t004:** Correlation of depressive symptoms, anxiety, and stress with exposure to infodemic of the elderly during the COVID-19 pandemic. Peru, 2021.

	Depressive Symptoms		Anxiety		Stress	
Variable	r *	*p* †	r	*p*	r	*p*
Internet	0.03	0.85	0.01	0.47	0.04	0.003
TV	0.01	0.22	0.03	0.03	0.03	0.03
Radio	0.03	0.03	0.03	0.01	–0.03	0.02

* r = correlation; † *p* = *p*-value.

**Table 5 ijerph-19-16893-t005:** Association between anxiety, stress, and depressive symptoms with demographic variables, infodemic, and physical symptoms in the elderly during the COVID-19 pandemic. Peru, 2021.

	Anxiety			Stress			Depressive Symptoms		
Variables	OR *	*p* †	CI 95%	OR	*p*	CI 95%	OR	*p*	CI 95%
Sex (female)	−0.16	0.03	0.73–0.98	-	-	-	-	-	-
Age	0.01	0.01	1.00–1.02	-	-	-	0.03	<0.001	1.02–1.04
Marital status (with a partner)	-	-	-	0.26	0.02	1.03–1.63	-	-	-
Internet exposure	-	-	-	0.03	0.04	1.00–1.06	-	-	-
TV exposure	-	-	-	0.03	0.02	1.00–1.07	-	-	-
Radio exposure	-	-	-	−0.05	<0.001	0.91–0.97	-	-	-
Headache (not)	0.24	0.03	1.02–1.58	-	-	-	-	-	-
Cold sweat/chills (not)	-	-	-	-	-	-	0.34	<0.001	1.19–1.66
Dry mouth (not)	-	-	-	-	-	-	0.18	0.04	1.00–1.43
Tremors (not)	0.52	<0.001	1.38–2.04	−0.47	<0.001	0.53–0.73	-	-	-
Muscle aches (not)	−0.23	0.03	0.63–0.97	0.43	<0.001	1.29–1.84	-	-	-
Chest tightness (not)	0.74	<0.001	1.73–2.54	-	-	-	0.42	<0.001	1.26–1.82
Difficulty breathing (not)	0.33	<0.001	1.16–168	-	-	-	0.29	0.001	1.12–1.60
Nutritional problems (not)	0.59	<0.001	1.49–2.18	-	-	-	0.40	<0.001	1.25–1.79
Palpitations (not)	0.37	<0.001	1.18–1.78	-	-	-	0.42	<0.001	1.26–1.84
Tiredness (not)	0.54	<0.001	1.36–2.16	-	-	-	0.27	0.006	1.08–1.61
Sleeping problems (not)	0.33	0.002	1.13–1.71	0.27	0.001	1.11-1.56	0.24	0.009	1.06–1.53

* OR = odds ratio; † *p* = *p*-value; ±CI = Confidence interval.

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
