# Peer review of "The Mental Health of the Peruvian Older Adult during the COVID-19 Pandemic"

_ijerph, 2022, doi:10.3390/ijerph192416893_

Round 1
Reviewer 1 Report
The Mental Health of the Peruvian Older Adult during the 2 COVID-19 Pandemic
The title did not relate to the introduction, where we did not have any context of Peruvian society during the covid pandemic; a slight change in the title or introduction is recommended.
Abstract: the conclusion presented in line 29/30 did not respond to the objective of the study
Keywords: question the keyword Peru since the introduction did not relate covid in that context
Introduction -
Sum: The introduction did not relate to the title, where we did not have any context of Peruvian society during the covid pandemic; changes in the introduction are recommended to compare the Peruvian reality with the rest of the world.
Line 37 . explain what restrictive measures were those.
Line 44 - Improving the formulation of the phrase is recommended in English.
Line 51 - no refs to the studies; recommended having those.
Line 68 - information on ethical approval for the study is needed
Line 97 - information on the scales and if they were adapted to the Peruvian population in the study is needed.
Lone 123 to 139 - information about the normalisation test of the sample is needed and to what tests were done.
Discussion
The discussion lack depth in the relation of the results and their consequences for the older generation.
The confrontation with the literature is still reduced; they can bring other international studies to the discussion to improve it.
Line 205 - none of the results of no married was reported for loneliness or fear of dying alone.
Line 231 to 237 - not sure this issue was well addressed, there are other reasons for these symptoms, and it recommended that this paragraph be reformulated.
Line 238 to 242 - Do the results presented relate to the general population? The low-income population already lacked nutrients, and the covid just increased it …this paragraph does not show the fundamental problems of access to older people to food. It recommended that a better explanation of the reality of older Peruvian people should be given.
Line 254 - the only limitation presented does not reflect the difficulties of the study; more should be pointed out
Conclusions: the use of media was not the primary goal of the study, and the conclusion should be reformulated to relate to the objectives presented above.
Author Response
The introduction was restructured according to the reviewer's suggestions and consistent with the purpose of the study.
The scales used were translated and validated into Spanish and, regardless of the population of the validated study, it was used because Peru still does not have the habit of validating instruments for the population, in addition to being a multicentric study, it is necessary to follow the indications of the coordinators .
in the discussion, the modifications and deepening of the results were carried out. As for the limitations of the study, the reviewer disagrees, since the limitations presented in the course of the research are clear.
The conclusions have been modified.

Reviewer 2 Report
Introduction
*During the pandemic, studies have been developed to know the characteristics of the virus, epidemiological aspects, and the different public health measures (References ??).
Comment: Please give reference for the statement
*However, there is still little research on the repercussion on mental health in the elderly (ref).
Comment: Please give reference for the statement . This paper can be a good reference for you :Mostert CM, Mackay D, Awiti A, Kumar M, Merali Z. Does social pension buy improved mental health and mortality outcomes for senior citizens? Evidence from South Africa's 2008 pension reform. Prev Med Rep. 2022 Oct 17;30:102026. doi: 10.1016/j.pmedr.2022.102026. PMID: 36310690; PMCID: PMC9596742.
*The significance of the study, the gap in the literature, and the unique contribution of this paper need to be separated into three theme paragraphs to enhance the quality of the introduction. Please consider such revision.
Methods
*I don't see any statement on the ethical consideration of your data collection strategy
* I don't see you addressing confounding factors that may be associated with the mental health of elderly population especially the socio-economic variables . This information is critical.
Results
*We need to see urban and rural split in your results--geographic location is a key determinant of mental health.
Discussion
* Your discussion lacks the policy implication of your study --a paragraph or two reflecting on this point is needed.
Conclusion
* There is a mismatch between your conclusion paragraph and the title of the paper . Please re-write this
Author Response
Modifications were made according to the reviewer and information about the ethics committee was added

Round 2
Reviewer 1 Report
If the questionnaire was not translated to Peruvian Spanish, this will bring issues of equivalence and should be reported as a limitation,
Author Response
Dear
The English revision of the entire text was carried out and the limitation of the use of the instrument was written.
Yours sincerely
